# GETTING TO THE CRUX: GRAPH-BASED DATA GENERATION FOR ADVANCING MULTI-HOP CROSS-MODAL REASONING

## ABSTRACT

Real-world reasoning often requires combining information across modalities—for example, following a recipe involves connecting textual instructions with visual cues in a multi-hop process. Yet, most multimodal benchmarks fail to capture this ability: they typically rely on single images or static sets, where answers can be inferred from one modality alone. This limitation is mirrored in training data, where interleaved image–text content rarely enforces complementary, multi-hop reasoning. As a result, Vision-Language Models (VLMs) frequently hallucinate and produce reasoning traces poorly grounded in visual evidence. To address this gap, we introduce CRUX, a new dataset and benchmark built with a scalable automatic pipeline for generating complex cross-modal reasoning tasks. CRUX spans natural images, videos, and text-rich sources, and includes a manually verified test set for reliable evaluation. Models trained on CRUX show significant gains in grounded multi-hop reasoning, including strong improvements on SPIQA and other multi-image benchmarks.

## 1 INTRODUCTION

As humans, we are constantly interacting with a multimodal world. Real-world tasks often involve reasoning over multiple, interleaved sources of information – for instance, when following a DIY tutorial or a recipe, we may find ourselves constantly cross-referencing textual instructions with a sequence of images (maybe even looking at a YouTube video in the middle!), engaging in a 'multi-hop' process to connect steps, tools, and outcomes (Chang et al., 2022; Luo et al., 2023; Cho et al., 2024). This requires grounding textual descriptions (e.g., 'fold the dough') to specific visual content (e.g., the actual visual stream of the dough that we see through our eyes) and tracking entities and states across this sequence.

Despite the importance of such tasks, most existing multimodal benchmarks do not adequately assess this capability. Current benchmarks, while valuable – typically present a single image (Masry et al., 2022; Liu et al., 2024; Yu et al., 2023; Chen et al., 2024b; Lu et al., 2023), a single video, or a static set of images (Wang et al., 2024a; Fu et al., 2024; Meng et al., 2024), with questions and annotation targets primarily in the text domain. Even in datasets designed for multi-stage reasoning (Zhang et al., 2024a; Nagrani et al., 2024; 2025), the necessary information can often be inferred from a single modality alone (eg. the video), failing to test true cross-modal grounding. In Figure 1, we provide an example of the complex reasoning we target. To arrive at the correct answer, a model must execute a multi-hop process: it must visually identify a 'gray object to the left of a banana', cross-reference text to find the 'year it was worn', use that year to find a related 'event' requiring approval, and finally link this back to another textual fact about the object's installation timeline. Crucially, the multimodal information is *complementary*; the visual data provides spatial and attribute information (shape, color, location) while the text provides temporal and abstract context (years, events) unavailable in the pixels alone.

This gap in evaluation is mirrored by a lack of suitable post-training or supervised fine-tuning (SFT) data. While a lot of interleaved image and textual data is used during training, it is unclear how much of it is truly complementary, and hence how much the model is forced to correlate the two modalitites. Given the scarcity of such complex interleaved data, it is perhaps unsurprising that

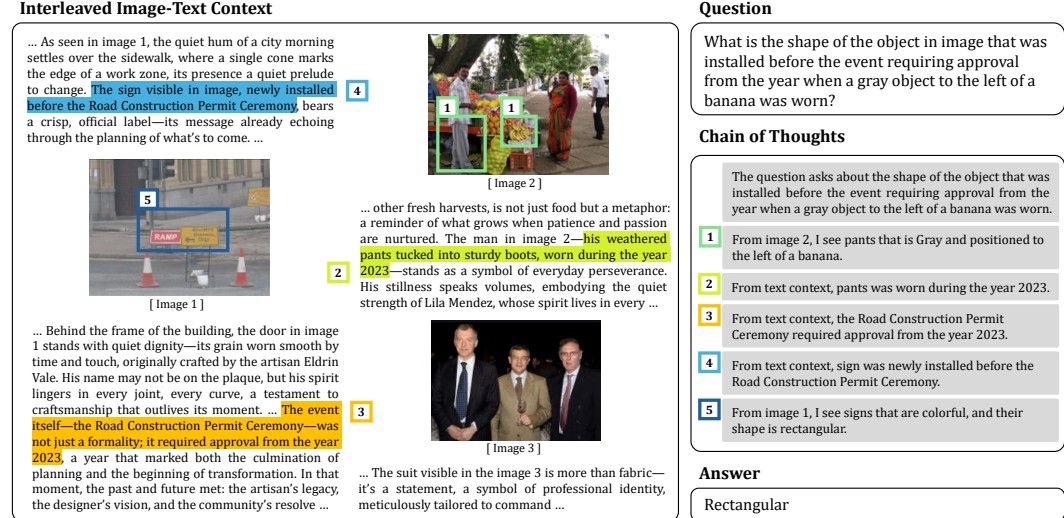

Figure 1: **Multi-hop, multi-modal reasoning.** We introduce a new image-text QA task, which requires complex, multi-hop reasoning to solve. This reasoning includes identifying and linking appropriate visual and textual elements from interleaved data to answer complex questions. We automatically collect a new dataset for this task called **CRUX** (**CR**oss-modal m**U**lti-hop reasoning over interleaved image-te**X**t).

existing Vision-Language Models (VLMs) struggle on such complex reasoning tasks. As we show in our experiments, when prompted for step-by-step reasoning, Chain-of-Thought (CoT) traces output by models are often poorly grounded in the visual evidence, frequently disjoint from the multimodal context and exhibiting significant hallucination.

In this work, we aim to address this problem by creating a scalable pipeline for generating high-quality, multi-hop, cross-modal reasoning data. Manually collecting such data is prohibitively expensive. While recent works have leveraged VLMs and LLMs to scale up data collection (Chen et al., 2024a; Guo et al., 2024; Huang et al., 2025; Xu et al., 2024; Yang et al., 2025), this approach has significant drawbacks for our target task. Tasking a VLM to automatically generate complex reasoning questions is prone to the same grounding failures and hallucinations we seek to measure. Furthermore, this risks creating cyclical biases, where models are evaluated on data generated by the very same class of models.

To overcome these challenges, we propose a novel graph-based automatic data generation pipeline for interleaved image-text content. Our pipeline is built on several key properties: (i) First, we use graphs as a structured representation of content, capturing entities, attributes, and relationships that appear in either modality. These are derived from clean and reliable sources, such as manually annotated grounded captions; (ii) Second, this structured format allows us to programmatically sample sub-graphs, guaranteeing the presence of complex, multi-hop relationships between modalities; and (ii) Finally, given a sampled sub-graph, we use a model to generate a complex question that necessitates multi-hop reasoning to be solved. By design, our pipeline does not require a VLM at any stage of the question-generation process (a text-only LLM is sufficient), thus avoiding the aforementioned cyclical biases and grounding issues.

Using this pipeline, we construct a novel dataset called **CRUX** (**CR**oss-modal m**U**lti-hop reasoning over interleaved image-te**X**t). Models trained on CRUX show improved cross-modal multi-hop reasoning, while our test set—built through our automatic pipeline and refined by human annotators—serves as a reliable benchmark for this capability.

To summarize, our key contributions are as follows:
1. We present an automatic data generation framework for cross-modal multi-hop reasoning across diverse domains, ranging from natural images to videos and text-rich sources such as scientific

papers. Our pipeline leverages graphs as a structured representation of multi-modal content.

2. We use this pipeline to create the CRUX dataset. We will release both a training and test set - the test set has been verified manually by human raters. Along with the test set, we also benchmark a number of state-of-the-art VLMs.

3. We further show that models trained on CRUX achieve significant improvements in cross-modal multi-hop reasoning, with notable improvements on SPIQA (Pramanick et al., 2024) and modest improvements on other multi-image benchmarks.

## 2 RELATED WORK

### 2.1 MULTIMODAL LARGE LANGUAGE MODELS

Multimodal large language models (MLLMs) have made notable progress in tasks that involve visual understanding and multi-step reasoning. However, their capabilities remain limited when the task requires reasoning across multiple modalities and multiple steps—what we refer to as cross-modal multi-hop reasoning (Chang et al., 2022; Talmor et al., 2021; Reddy et al., 2022; Zhao et al., 2024; Abaskohi et al., 2024; Wang et al., 2024b; Kim et al., 2024; Akhtar et al., 2025; Foroutan et al., 2025). In such tasks, the model must integrate information from both images and text, sometimes across several hops, to draw correct inferences . This limitation becomes especially apparent in settings where inputs are interleaved sequences of text and images, and successful reasoning depends on identifying and linking the appropriate visual and textual elements based on complex instructions (Zhou et al., 2024). A key reason for this shortcoming lies in the structure of existing training datasets, which typically emphasize either isolated visual content or shallow text-image pairs, lacking the deeper compositional reasoning required for multi-hop inference. A recent trend involves using MLLMs to synthesize multimodal training data (Guo et al., 2024; Chen et al., 2024a; Shi et al., 2024; Zhang et al., 2024b), but this approach often suffers from errors—especially in visual perception and hallucination. Moreover, most existing datasets lack explicit reasoning traces and consist solely of question-answer pairs, providing limited supervision for step-by-step reasoning.

### 2.2 INTERLEAVED IMAGE-TEXT DATASETS

Interleaved image–text datasets have recently become a standard choice for pretraining VLMs (Zhu et al., 2023; Laurençon et al., 2023; Li et al., 2024b; Zhang et al., 2025). However, large-scale interleaved multimodal datasets were developed primarily for pretraining and typically lack precise alignment between text and images, making them unsuitable as a foundation for cross-modal, multi-hop reasoning datasets. While pretraining on such data improves general multimodal ability, it does not guarantee strong performance on tasks requiring fine-grained interleaved understanding. VEGA (Zhou et al., 2024) targets interleaved image-text comprehension by constructing interleaved contexts from SciGraphQA (Li & Tajbakhsh, 2023), inserting a relevant image–text pair needed for answering a question and appending additional content. Yet this approach simplifies cross-modal interactions. In practice, real-world scenarios involve richer and more complex interleaving, where multiple images and passages interact in non-trivial ways. Designing datasets that move beyond weakly aligned interleaved corpora and capture these complexities is essential for advancing multimodal reasoning.

## 3 DATA GENERATION FOR CROSS-MODAL MULTI-HOP REASONING

In this work we explore cross-modal multi-hop reasoning using VLMs, as shown in Fig. 1. Obtaining suitable data for this task is non-trivial – even prior to annotation, the first challenge is identifying suitable source data, which contains high-quality image–text interleaved content with rich multimodal interactions. Once such content is collected, annotation requires identifying connected facts across modalities, formulating multi-hop questions and answers based on these facts, and ensuring that the supporting facts lead to a unique answer.

To address this challenge, we introduce a novel graph-based automatic data generation framework. Our framework operates in three stages, illustrated in Figure 2: **(1) constructing multimodal content graphs**, **(2) generating textual context** to accompany visual content, and **(3) generating question–answer pairs** that require cross-modal multi-hop reasoning based on the multimodal content

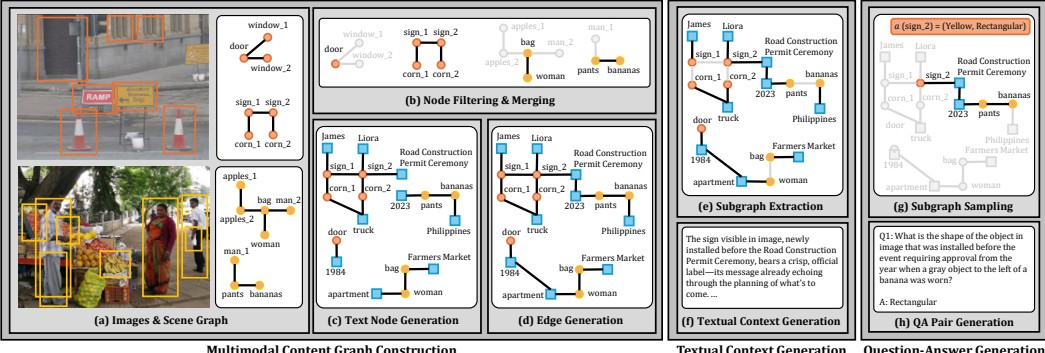

Figure 2: **Overall Process of Cross-Modal Multi-Hop QA Generation.** The procedure consists of three main stages. Multimodal Content Graph Construction: images annotated with scene graphs are sampled (a), unique entities are extracted and merged (b), and entities with relationships are generated via LLM prompting (c, d). Textual Context Generation: subgraphs are extracted from the multimodal content graph (e) to produce complementary textual descriptions (f). Question–Answer generation: subgraphs are further sampled (g) to enable cross-modal multi-hop reasoning, from which QA pairs are generated (h).

graph. In the following subsections, we describe each stage in detail. We first describe how our framework works with images that already contain annotated scene graphs, and then describe extensions to our pipeline that enable us to use videos and scientific papers as well (sec. 3.4).

## 3.1 MULTIMODAL CONTENT GRAPH CONSTRUCTION

The multimodal content graph lies at the core of our framework. It provides a structured representation of interleaved image–text content, consisting of entities, attributes, and relationships that appear in either modality. The graph is designed to encode both intra-modal and cross-modal relationships.

**Formulation of Multimodal Content Graphs.** We model visual and textual content in interleaved image–text as a directed graph $G = (\mathcal{V}, \mathcal{E})$. Each node $v \in \mathcal{V}$ represents an entity, either a visual object in an image or a textual entity. An edge $e \in \mathcal{E}$ between nodes $u$ and $v$ is represented as a triple $(u, v, r)$, where $r$ denotes the relation between the two entities within the multimodal content. The graph is further augmented with an attribute function $a(u)$ that returns the set of attributes associated with entity $u$. Finally, a modality function $m(\cdot)$ specifies the modality index: for an entity or relation, $m(\cdot) = i$ indicates presence in image $i$, while $m(\cdot) = 0$ denotes presence in the text.

**Graph Construction through Content Augmentation.** We begin with a dataset of images annotated with scene graphs, which eliminates the need for image generation or potential errors in scene-graph recognition. Since our target is interleaved image–text content involving multiple images, we randomly sample between one and six images from the annotated dataset (see Figure 2a). For each image, we retain only entities that can be uniquely identified by an attribute or by their relation to another entity, ensuring that each attribute or relation corresponds to exactly one entity (Figure 2b). This avoids ambiguity during the QA generation phase, where multiple valid answers could otherwise arise. The scene graphs of the sampled images are then merged into a single content graph, which serves as the starting point for our framework. To augment this graph with textual information, we leverage an LLM to add new entities and relations. Specifically, for each image node in the graph, we prompt the LLM with the full current graph and ask it to generate a plausible entity and relation connected to that node (Figure 2c). We then apply an additional round of prompting to create plausible relations among the newly added entities, which serve as bridges across different visual sources (Figure 2d). The prompts used are provided in Appendix A.2.1. These augmented entities and relations constitute the textual content; since they are not tied to a specific image, they naturally serve as bridges that connect entities across different images through related textual nodes. The resulting graph provides a coherent multimodal context for evaluating cross-modal and multi-hop reasoning, as it connects concepts drawn from multiple images and textual segments.

## 3.2 TEXTUAL CONTEXT GENERATION

After graph construction, each sample is represented by a unified graph structure, comprising both image and text nodes. For each image, we prompt the LLM with a subgraph to generate complementary textual context. The prompts used are provided in Appendix A.2.2. We begin extracting the subgraph that includes the corresponding image nodes and all text nodes directly connected to them, along with the edges between those text nodes. When an edge connects two text nodes that belong to different images, we handle it by randomly selecting one of the images and expanding its subgraph to include that edge. Including these edges is essential because they provide the only connections across independent images, enabling the construction of multi-hop reasoning questions and supporting cross-modal reasoning that spans multiple images. Attributes of image nodes and inter-image relations are excluded, as these are intended to be inferred from the visual modality during cross-modal reasoning. The generated passages therefore describe only the augmented text nodes, relationship between them and their connections to the image nodes. Diverse narrative styles—such as stories, diary entries, or documentary-style scripts are used that provide linguistic variety. By combining the generated texts with the images, the framework yields complementary multimodal content that facilitates cross-modal multi-hop reasoning.

## 3.3 QUESTION-ANSWER PAIR GENERATION

Once the multimodal content is generated by augmenting images with text, we proceed to generate cross-modal multi-hop QA pairs. To this end, we sample a subgraph from the multimodal content graph that represents a chain of facts within the content. Since our goal is cross-modal reasoning, we ensure that the sampled subgraph includes nodes from both images and text. To guarantee multi-hop reasoning, we further restrict the subgraphs to have $1 \leq h \leq 5$ edges, such that answering the resulting questions requires reasoning over multiple connected facts. Finally, we require the terminal node in each chain to come from an image. We then select either its entity name or one of its attributes as the answer, grounding the final reasoning step in the image content. This strategy improves the reliability of evaluation by preventing the model from guessing answers based solely on textual biases or hallucinations without performing the correct reasoning. If the terminal node's immediate neighbor is textual, we instead use its attribute as the final answer, since the entity name is already present in the text context.

Given a sampled subgraph (serialized as JSON) and its designated answer, we prompt the LLM to generate a question under the following constraints: (i) the answer to the generated question must match the provided target answer; (ii) solving the question must require reasoning over the entire chain of facts in the subgraph, thereby enforcing cross-modal multi-hop reasoning; and (iii) intermediate entities should not be mentioned explicitly in the question but instead be recoverable only through multi-hop inference. The prompts used are provided in Appendix A.2.3. While the QA pair provides the basis for evaluating cross-modal multi-hop reasoning, we additionally generate a CoT trace (Wei et al., 2022) to guide the model through the step-by-step reasoning process. Since the subgraph corresponding to each QA already encodes the complete reasoning chain, we can leverage it to prompt the LLM to produce a CoT response for the generated question, as in Appendix A.2.4. This offers extra supervision by explicitly indicating where each piece of information should be retrieved from within the multimodal context.

Once the QA samples with CoT are generated, we apply an LLM-based staged filtering process to ensure quality. First, we discard questions that explicitly mention intermediate entities from the sampled subgraph, thereby ensuring multi-hop reasoning by eliminating potential shortcuts in the reasoning chain. Next, we filter out questions that can be answered using a single modality. To perform this test, we provide off-the-shelf LLMs with the nodes and edges corresponding to each modality; Since visual content is translated into structured text through the graph representation, a text-only LLM is sufficient. We use three different LLMs and remove a question if all three predict the correct answer using only one modality (either text or visual). Finally, we prune CoT responses that are excessively verbose by limiting the output to a maximum of ten sentences.

## 3.4 EXTENSIONS TO VIDEOS AND TEXT-RICH SOURCES

The data creation pipeline described above is limited in two aspects: (i) first, it relies on images that have graph annotations, and (ii) the images are independent and unrelated to one another, as well as

being limited to the natural domain. We would also like to explore text-rich visuals such as tables, diagrams, and charts that frequently occur alongside textual context in real-world documents  (Jia et al., 2024; Hu et al., 2024; Zhang et al., 2023). To address these gaps, we extend our pipeline to two additional data sources: (i) still frames from videos, where multiple coherent frames naturally share contextual information, and (ii) scientific papers, which provide abundant text-rich images.

For video frames, the main challenge is the absence of scene-graph annotations. To address this, we leverage dense video captioning datasets, where long-form videos are annotated with temporally localized event descriptions. We select the frames with the highest CLIP similarity to each caption to ensure that the visual content aligns closely with the description. Although these captions may be short, they provide sufficient content information to be converted into partial scene graphs—similar to the manual scene graph annotation process in  (Krishna et al., 2017b), where captions are collected and then transformed into graphs. Concretely, we prompt an LLM to convert these descriptions into graphs using the prompt detailed in Appendix A.2.5. Sampled frames from a single video may share entities and relations, and therefore we incorporate this into both the prompting design and the resulting graphs. After constructing a scene graph across coherent frames, the rest of the pipeline proceeds as before, except that textual context is now generated once for the entire image set rather than separately for each image.

In the case of text-rich images, scientific papers inherently provide high-quality interleaved image–text content. Because well-aligned textual descriptions already exist, generating complementary textual context is unnecessary. Nonetheless, constructing QA pairs would benefit from a graph-based representation. To this end, we convert the multimodal content—paragraphs, figures, and tables—into a unified graph structure, treating both figures and rendered tables as images. Given a paper containing textual content with associated figures and tables, we first construct a content graph from paragraphs that do not reference any visual elements using LLM, with prompt detailed in Appendix A.2.6. For paragraphs that reference figures or tables, we prompt the LLM to identify visual entities and relations, detailed in Appendix A.2.7. Numerical entities, comparisons, and explicit visual descriptions in these paragraphs are elevated to visual nodes in the graph. To enforce cross-modal reasoning, the corresponding sentences are then removed from the text body once they are marked as visual. Finally, once the corresponding multimodal content graph is constructed, QA generation is performed following the same base pipeline.

## 4 CRUX

In this section, we describe the implementation details used to create CRUX (sec. 4.1), and then describe dataset statistics (sec. 4.2).

### 4.1 Dataset Creation Implementation Details

**Data Sources**  Most of the dataset comes from GQA (Hudson & Manning, 2019), which provides *natural images* annotated with scene graphs that were sampled and incorporated following the procedure from Section 3.1 to Section 3.3. Beyond these, CRUX was additionally constructed from *video frames* and captions obtained from the dense video captioning dataset ActivityNet Captions (Krishna et al., 2017a), as well as high-quality interleaved image–text data from *scientific papers* is obtained from SPIQA (Pramanick et al., 2024).

**Models**  Qwen3-30B-A3B-Instruct-2507 was used at every stage of the data generation pipeline. For the final filtering of questions and answers, we used Qwen3-30B-A3B-Instruct-2507, Gemma-3-27b-it and Mistral-Small-3.2-24B-Instruct. (Team, 2025a;b; AI, 2025)

**Human Verification**  For the test set, we recruited 13 English-proficient undergraduate students majoring in STEM fields to filter out samples that did not adhere to the annotation guidelines. Detailed guidelines are provided in Appendix A.1.

### 4.2 Dataset Statistics

CRUX consists of samples, which are unique image sets. Each sample consists of multiple QA pairs per image set, concatenated into a multi-turn conversation. In total, CRUX consists of 84,179

Table 1: **CRUX Data Statistics.** The dataset consists of natural images, video frames and scientific papers. We report the number of samples and QA pairs, with the proportion of QA pairs involving image–image reasoning (in addition to image–text reasoning) shown in brackets.

| Split | Statistic | Natural Images | Videos | Scientific Papers |
|---|---|---|---|---|
| Train | # of Samples | 49,151 | 8,010 | 27,018 |
| | Avg. Images per Sample | 3.8 | 3.5 | 3.7 |
| | Avg. Text Tokens per Sample | 2,442 | 649 | 5,443 |
| | # of QA | 152,368 (0.0%) | 16,020 (21.6%) | 81,054 (6.1%) |
| Test | # of Samples | 990 | 583 | 415 |
| | Avg. Images per Sample | 4.4 | 5.2 | 7.7 |
| | Avg. Text Tokens per Sample | 2,930 | 802 | 5,338 |
| | # of QA | 990 (0.0%) | 583 (34.0%) | 498 (5.3%) |

Table 2: **Results on CRUX.** Comparison of proprietary, open-source, and fine-tuned multimodal language models across natural images, videos, and scientific papers. Models Fine-tuned with CRUX are highlighted in bold with grey background. EM: exact match, Ref. Acc: reference accuracy.

| Model | #Param | Natural Image | | | Video | | | Scientific Paper | | |
|---|---|---|---|---|---|---|---|---|---|---|
| | | EM | F1 | Ref. Acc | EM | F1 | Ref. Acc | EM | F1 | Ref. Acc |
| *Proprietary Models* | | | | | | | | | | |
| GPT-4o | - | 24.0 | 26.6 | 94.4 | 11.7 | 16.3 | 72.2 | 5.1 | 11.0 | 56.5 |
| GPT-4o-mini | - | 23.7 | 25.7 | **94.8** | 12.4 | 16.7 | 70.5 | 4.9 | 10.2 | 55.3 |
| *Open-Source Models* | | | | | | | | | | |
| Phi3.5-Vision | 4B | 7.9 | 8.5 | 62.16 | 10.1 | 12.2 | 38.4 | 3.3 | 6.5 | 1.0 |
| Qwen2.5-VL (7B) | 7B | 21.2 | 22.0 | 57.5 | 8.6 | 9.9 | 39.5 | 7.3 | 8.8 | 24.6 |
| **Qwen2.5-VL**$_{CRUX}$ | 7B | **52.3** | **52.7** | 93.4 | **35.0** | **37.6** | 44.3 | **12.8** | **18.9** | 40.8 |
| LLaVA-Onevision | 7B | 5.8 | 19.6 | 0.0 | 18.4 | 24.0 | 0.0 | 1.8 | 8.2 | 0.0 |
| InternVL2.5 | 8B | 20.3 | 21.3 | 49.0 | 13.9 | 17.8 | 12.4 | 4.2 | 7.6 | 12.6 |
| Idefics2 | 8B | 15.4 | 15.9 | 1.5 | 19.7 | 22.3 | 2.7 | 2.4 | 5.2 | 0.2 |
| **Idefics2**$_{CRUX}$ | 8B | 50.8 | 51.0 | 92.5 | 22.8 | 25.4 | 40.1 | 12.4 | 18.6 | 25.7 |
| Qwen2.5-VL (72B) | 72B | 26.6 | 27.8 | 90.2 | 16.1 | 18.3 | **76.9** | 7.7 | 9.9 | **59.2** |

samples in training and 1,988 in testing, and 249,442 QA pairs in the training set and 2,207 QA pairs in the test set. Table 1 presents the statistics for each data source. Note how the majority of QA pairs are generated from natural images annotated with scene graphs. Some QA pairs also include image-image reasoning in addition to image–text reasoning. This proportion varies across the different domains. For natural images consisting of independent images, this proportion is 0% (as none of the images are related). For videos composed of sequential frames, the proportion is highest, at 21.6% in the training set and 34.0% in the test set. For scientific papers, the corresponding proportions are 6.1% and 5.3%.

## 5 EXPERIMENTS

### 5.1 EXPERIMENTAL SETUP

**Base Models** We evaluate both open-source and proprietary models. For open-source models, we evaluate Qwen2.5-VL (Bai et al., 2025), LLaVA-Onevision (Li et al., 2024a), Idefics2 (Laurençon et al., 2024), Intern2.5-VL (Chen et al., 2024c), Phi3.5-Vision (Abdin et al., 2024). For proprietary models we use GPT-4o and GPT-4o-mini (Hurst et al., 2024).

**Training Settings** We perform supervised fine-tuning with LoRA (Hu et al., 2022) on multi-image vision-language models, specifically Qwen2.5-VL-Instruct and Idefics2-8B. Each fine-tuned models are notated as Qwen2.5-VL$_{CRUX}$ and Idefics2$_{CRUX}$. Training is conducted on CRUX using a mixture

Table 3: **Performance on SPIQA with direct QA and CoT.** Results of Idefics2 models fine-tuned on Mantis instruction tuning set and additional CRUX training set. We measure performance both in direct answer and answer with CoT trace. Model fine-tuned with dataset including CRUX shows remarkable performance improvement. Metrics include M: METEOR, R-L: ROUGE-L, C: CIDEr, B-F1: BERTScore F1, and Ret: Retrieval accuracy.

| Model | Train Data | CoT | SPIQA test-A | | | | | SPIQA test-B | | | | | SPIQA test-C | | | | |
|---|---|---|---|---|---|---|---|---|---|---|---|---|---|---|---|---|---|
| | | | Ret | M | R-L | C | B-F1 | Ret | M | R-L | C | B-F1 | Ret | M | R-L | C | B-F1 |
| Idefics2 | – | ✗ | - | 2.3 | 2.1 | 2.6 | 28.3 | - | 0.8 | 1.1 | 0.4 | 25.8 | - | 0.3 | 0.1 | 0.1 | 30.6 |
| Idefics2 | – | ✓ | 8.7 | 9.1 | 15.4 | 26.4 | 47.8 | 12.7 | 4.2 | 10.3 | 13.8 | 43.2 | 14.2 | 3.0 | 6.0 | 9.5 | 38.8 |
| Idefics2 | Mantis-Inst. | ✗ | - | 6.5 | 14.4 | 43.4 | 45.7 | - | 2.7 | 7.3 | 9.6 | 39.6 | - | 1.6 | 8.8 | 18.5 | 43.2 |
| Idefics2 | Mantis-Inst. | ✓ | 37.1 | 5.6 | 7.7 | 9.0 | 41.9 | 29.4 | 2.7 | 5.3 | 4.4 | 39.8 | 37.7 | 2.6 | 3.2 | 14.0 | 37.2 |
| Idefics2 | Mantis-Inst. + CRUX | ✗ | - | **19.0** | **33.3** | **127.4** | **63.1** | - | 6.9 | **16.6** | **31.4** | **50.9** | - | **5.7** | **18.4** | **45.0** | **53.4** |
| Idefics2 | Mantis-Inst. + CRUX | ✓ | **58.9** | 16.3 | 24.1 | 50.7 | 55.9 | **32.5** | **7.0** | 11.3 | 5.0 | 46.6 | **44.0** | 4.5 | 12.1 | 25.4 | 43.5 |

Table 4: **Performance on other benchmarks.** Models trained with our dataset achieve performance that is consistently higher or at least comparable across all benchmarks, compared to those trained without it.

| Model | Train Data | FCMR | MuirBench | BLINK | MP-DocVQA |
|---|---|---|---|---|---|
| Idefics2 | - | 40.5 | 26.2 | 45.2 | 46.7 |
| Idefics2 | Mantis-Inst. | 44.9 | **33.1** | 45.7 | 48.2 |
| Idefics2 | Mantis-Inst. + CRUX | **50.5** | 32.6 | **47.6** | **49.8** |

of direct-answer and chain-of-thought (CoT) responses, so a single model is trained jointly on both data types.

**Evaluation Details** Evaluation spans natural images, video, and scientific papers, with Exact Match (EM) and F1 as metrics. We also report *Reference Accuracy*, measuring whether the model correctly identified the necessary images to answer the question. All prompts use CoT, and models are instructed to explicitly reference the relevant image.

## 5.2 RESULTS ON CRUX

Table 2 reports results across natural images, videos, and scientific papers. Open-source models without fine-tuning perform poorly. They often fail to localize the relevant image or, even when successful, provide incorrect answers due to visual perception errors or the inability to connect information across modalities. Fine-tuning on CRUX leads to substantial gains across all domains. Qwen2.5-VL$_{\text{CRUX}}$ achieves the best EM and F1 score for natural images, its reference accuracy reaching the level of proprietary models. Idefics2$_{\text{CRUX}}$ which didn't have the capability to find the relevant sources to answer the question improved substantially. The relatively high scores in the video domain for LLaVA-OneVision and Idefics2 were obtained while they were instructed to engage in reasoning, they provided short answers that fortunately turned out correct.

## 5.3 RESULTS ON OTHER BENCHMARK

While the CRUX results are encouraging, a central question is whether fine-tuning on CRUX induces genuine cross-modal reasoning ability, and how CRUX-style data impacts performance when incorporated into large-scale instruction tuning. To investigate this, we evaluate on Mantis-Instruct (Jiang et al., 2024), a dataset of 721K multi-image instruction–response pairs. We compare the zero-shot performance of Idefics2 under two settings: (i) fine-tuned solely on Mantis-Instruct, and (ii) fine-tuned jointly on Mantis-Instruct and CRUX. We evaluate on benchmarks that For evaluation we test SPIQA (Pramanick et al., 2024), a scientific open-ended QA dataset where models are given multiple figures and tables along with its corresponding captions, and the model needed to identify which image to refer at. We also evaluate on FCMR (Kim et al., 2024), a financial cross-modal multi-hop benchmark where the model needs to jointly reason over text, table and chart. MuirBench (Wang et al., 2024a) and BLINK (Fu et al., 2024) are used to test multi-image visual perception. MP-DocVQA (Tito et al., 2023) to test multi-page document understanding. For evaluation, we consider diverse benchmarks spanning multiple domains. SPIQA (Pramanick et al., 2024) assesses scientific

Table 5: **Impact of Data Scaling.** Results of fine-tuning on the original CRUX training set and on the CRUX training set augmented with off-the-shelf models. Performance is reported on the CRUX test set across three domains: natural images, videos, and scientific papers. EM: exact match, Ref. Acc: reference accuracy.

| Model | Finetuned | Natural Image | | | Video | | | Scientific Paper | | |
|---|---|---|---|---|---|---|---|---|---|---|
| | | EM | F1 | Ref Acc | EM | F1 | Ref Acc | EM | F1 | Ref Acc |
| Qwen2.5-VL-7B | - | 21.2 | 22.0 | 57.5 | 8.6 | 9.9 | 39.5 | 7.3 | 8.8 | 24.6 |
| Qwen2.5-VL-7B | CRUX | 52.3 | 52.7 | 93.4 | 35.0 | 37.6 | 44.3 | 12.8 | 18.9 | 40.8 |
| Qwen2.5-VL-7B | CRUX Augmented | **57.6** | **58.1** | **95.0** | 32.1 | 35.1 | **44.6** | **12.8** | **19.4** | **41.7** |

open-ended QA, where models must select the relevant figures or tables given multiple candidates along with its caption—a setting closely aligned with our task. FCMR (Kim et al., 2024) measures financial cross-modal multi-hop reasoning across text, tables, and charts. MuirBench (Wang et al., 2024a) and BLINK (Fu et al., 2024) focus on multi-image visual perception, while MP-DocVQA (Tito et al., 2023) evaluates multi-page document understanding. Evaluation metrics vary across tasks: n-gram based metrics and retrieval accuracy (aligned with CRUX reference accuracy) are used for SPIQA, F1 score for FCMR, and accuracy for the remaining benchmarks.

The experimental results indicate that incorporating CRUX yields consistent performance improvements. As shown in Table 3, models fine-tuned with CRUX generate responses that more closely align with ground-truth references and exhibit substantial gains in retrieval accuracy, exceeding 20 points on the test-A split. These findings highlight CRUX's effectiveness in enabling models to identify and reason over relevant information in interleaved multimodal contexts. Table 4 further demonstrates generalization across benchmarks. On FCMR, despite CRUX not being explicitly designed for multi-hop financial reasoning, we observe marked gains in F1. For multi-image perception, performance improves on BLINK, with only a minor drop on MuirBench. Finally, CRUX enhances MP-DocVQA results, underscoring improved reasoning over long, multi-page documents.

### 5.4 IMPACT OF DATA SCALING

Furthermore, we demonstrate the effectiveness of automatic data augmentation using off-the-shelf models in enhancing cross-modal multi-hop reasoning capabilities. We augment the CRUX training set based on natural images. Specifically, we generate scene graphs with the scene graph generation model EGTR (Im et al., 2024) and generate object attributes with Qwen2.5-VL. Through this process, we construct additional 100K QA pairs. The training results are shown in Table 5. Fine-tuning with augmented CRUX leads to substantial performance gains, showing approximately 164% and 10% relative improvements on the F1 score for natural images compared to the non–fine-tuned model and the model fine-tuned on CRUX, respectively. For scientific papers, the augmented CRUX also achieves relative improvements of approximately 120% and 3% compared to the non–fine-tuned model and the model fine-tuned on CRUX, respectively. However, the effect of data augmentation is limited for video domain. We conjecture that this is due to the difference in data characteristics: while natural images consist of independent images, video still frames are composed of sequential images. Moreover, the distribution differs because the number of QA pairs generated from natural images is substantially larger, which we believe contributed to this outcome. Despite these limitations, data augmentation with model-generated data demonstrates scalable and robust improvements, highlighting the strength of our pipeline for enriching multimodal reasoning tasks.

## 6 CONCLUSION

We explored the challenge of cross-modal multi-hop reasoning, where solving a task requires weaving together complementary evidence from interleaved text and images. To advance this capability, we introduced CRUX, built through a novel graph-based pipeline that yields scalable, high-quality training and evaluation data. Fine-tuning on CRUX delivers notable gains across diverse benchmarks, and our augmentation experiments demonstrate a practical path to further scaling. We believe CRUX provides a foundation for developing models with more robust and grounded multimodal reasoning.

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

## A  APPENDIX

### A.1  HUMAN VERIFICATION GUIDELINE

For each sample, raters are provided with:

- One or more **images** (e.g., natural photographs, video frames, or figures from research papers).
- A **textual context** (such as generated descriptions or filtered text).
- A **question** and its corresponding **answer**.
- The **subgraph** used to generate the question–answer pair.

#### A.1.1  GUIDELINE FOR TASK

The task of the rater is to judge whether the given question–answer pair constitutes a valid example of cross-modal multi-hop reasoning. Each sample must be classified into one of three categories:

- **Keep**: Valid example that should be included in the benchmark.
- **Discard**: Invalid example that should not be included in the benchmark.
- **Unsure**: Uncertain case that requires further review or discussion.

#### A.1.2  CHECKLIST FOR DECISION-MAKING

To ensure consistency, the raters receive a checklist to decide whether to **Keep** or **Discard** a sample.

**Keep the sample if:**

- The question can only be answered by integrating **both image(s) and text**.
- The reasoning requires **multi-hop inference** (chaining across multiple entities, edges, or modalities).
- The answer is correct, unambiguous, and consistent with the given evidence.
- The question is natural, clear, and does not contain awkward phrasing or hallucinations.
- There is only one answer and no other alternatives that fit the evidence.

**Discard the sample if:**

- The question can be answered using **only text** or **only image(s)** without cross-modal reasoning.
- The reasoning does not actually require **multi-hop** (i.e., a single entity lookup is enough).
- The answer is incorrect, incomplete, or contradictory.
- The question is **ill-posed** (ambiguous, nonsensical, ungrammatical, or explicitly revealing the reasoning trace).
- The question ignores information from the subgraph or incorrectly conveys relationships.
- There are multiple valid answers or the answer is subjective.

## A.2 Prompts for Data Generation

### A.2.1 Graph Construction Prompt

We employ six types of Text Node Generation Prompts. Below, we present three representative examples. In addition, we include the Edge Generation Prompt.

---

**Text Node Generation Prompt Type 1**

```
You are generating a fact about an object or entity that appears in an image.

Inputs you will receive:
- Object: the target entity.
- Image Caption: a one-sentence description of the whole image involving the object/entity.
- Object Caption: a short description focusing specifically on the object/entity.

Task:
Generate exactly one NEW fact about the object/entity in the category: Authorship / Creation
/ Discovery.

Rules:
- The fact must be NON-VISUAL (cannot be inferred from appearance or caption).
- The fact must be NON-COMMONSENSE (not universally true or obvious).
- Do not contradict either caption.
- Avoid mythical, fantasy, or obviously fictional names, rituals, or events.
- Names can be synthetic but should sound plausible.

Output format (JSON only):
{{
"subject": "<the given object/entity>",
"relation": "<the relation type>",
"object": "<the new entity, formatted as 'type (name)'>"
}}

Examples:
- {{"subject": "chair", "relation": "designed by", "object": "artisan (Liora Vex)"}}
- {{"subject": "compass", "relation": "invented by", "object": "engineer (Tavian Sorrell)"}}
- {{"subject": "man", "relation": "discovered", "object": "invention (Quantum Lens)"}}

Object: {object}
Image Caption: {image_caption}
Object Caption: {object_caption}
```

Figure 3: Text Node Generation Prompt Type 1.

**Text Node Generation Prompt Type 2**

```
You are generating a fact about an object or entity that appears in an image.

Inputs you will receive:
- Object: the target entity.
- Image Caption: a one-sentence description of the whole image involving the object/entity.
- Object Caption: a short description focusing specifically on the object/entity.

Task:
Generate exactly one NEW fact about the object/entity in the category:
Human Involvement / Institutional Association.

Rules:
- The fact must be NON-VISUAL (cannot be inferred from appearance or caption).
- The fact must be NON-COMMONSENSE (not universally true or obvious).
- Do not contradict either caption.
- Avoid mythical, fantasy, or obviously fictional names, rituals, or events.
- Names can be synthetic but should sound plausible.

Output format (JSON only):
{{
"subject": "<the given object>",
"relation": "<the relation type>",
"object": "<the new entity, formatted as 'type (name)'>"
}}

Examples:
- {{"subject: "man", "relation": "employed by", "object": "company (TechNova)"}}
- {{"subject: "boy", "relation": "friend of", "object": "person (Elias Thorn)"}}

Object: {object}
Image Caption: {image_caption}
Object Caption: {object_caption}
```

Figure 4: Text Node Generation Prompt Type 2.

**Text Node Generation Prompt Type 3**

```
You are generating a fact about an object or entity that appears in an image.

Inputs you will receive:
- Object: the target entity.
- Image Caption: a one-sentence description of the whole image involving the object/entity.
- Object Caption: a short description focusing specifically on the object/entity.

Task:
Generate exactly one NEW fact about the object/entity in the category:
Temporal / Historical Facts.

Rules:
- The fact must be NON-VISUAL (cannot be inferred from appearance or caption).
- The fact must be NON-COMMONSENSE (not universally true or obvious).
- Do not contradict either caption.
- Avoid mythical, fantasy, or obviously fictional names, rituals, or events.
- Names can be synthetic but should sound plausible.

For years/ages, make them synthetic but plausible (e.g., "year (2005)", "4 years").
Every object exists in the current year so the year or age should not be out of
a reasonable range.

Output format (JSON only):
{{
"subject": "<the given object>",
"relation": "<the relation type>",
"object": "<the new entity, formatted as 'type (name)'>"
}}

Examples:
- {{"subject": "car", "relation": "manufactured in", "object": "year (2005)"}}
- {{"subject": "dog", "relation": "has age", "object": "4 years"}}
- {{"subject": "spoon", "relation": "made in", "object": "year (2010)"}}

Object: {object}
Image Caption: {image_caption}
Object Caption: {object_caption}
```

Figure 5: Text Node Generation Prompt Type 3.

**Edge Generation Prompt**

```
You are given a list of entities in the format "type (name)".
Your task is to generate plausible synthetic relations between these entities.

### Rules:
- Each output must be a JSON object with the format:
{{
    "subject": "<entity from the list>",
    "relation": "<synthetic relation connecting it to another entity>",
    "object": "<entity from the list>"
}}

- The 'subject' and 'object'' must always come from the given list.
- Relations must be plausible. If there is no reasonable relation, the output should be
an empty list.
- Do not invent new entities outside of the given list.
- Output only a list of JSON objects (no extra text).

### Example:
Input:
["institution (Museum of Oracles)", "event (Expo 2020)", "artifact (Singing Blade)",
"concept (Eternal Silence)"]

Output:
[
    {{"subject": "institution (Museum of Oracles)", "relation": "preserves",
      "object": "artifact (Singing Blade)"}},
    {{"subject": "concept (Eternal Silence)", "relation": "inspires",
      "object": "artifact (Singing Blade)"}},
    {{"subject": "event (Expo 2020)", "relation": "hosts",
      "object": "institution (Museum of Oracles)"}},
    {{"subject": "artifact (Singing Blade)", "relation": "represents",
      "object": "concept (Eternal Silence)"}},
    {{"subject": "institution (Museum of Oracles)", "relation": "exhibits",
      "object": "event (Expo 2020)"}},
]

### Input:
{list_of_entities}

### Output:
```

Figure 6: Edge Generation Prompt.

### A.2.2 TEXTUAL CONTEXT GENERATION PROMPT

For the Textual Context Generation Prompt, we define a set of diverse context types. These include Story/Narrative, Newspaper Article, Comedy Sketch, Diary Entry, Poem, Song Lyrics, Documentary Script, Blog Post, Motivational Speech, Promotional Article, Movie Scene Description, and Social Media Post. We randomly use one of them in each prompt.

```
Textual Context Generation Prompt

You are writing a {context_type}.

The following entities and relations must be included:

Entities:
[Entities list]

Relations:
[Relations list]

Detailed Guidelines:
1. Explicit Image References
- Every entity that contains "(Image N)" MUST be explicitly tied to its image number in the
text. Do this by phrases like "as seen in image N", "shown in image N", or "visible in
image N".
- Every entity that contains "(Image)" MUST be described as appearing in that image. Do this
by phrases like "as seen in the image", "shown in the image", or "visible in the image".
- Example: Instead of writing "The telephone pole is maintained by Veridian Grid Solutions",
write "The telephone pole shown in image 1 is maintained by Veridian Grid Solutions."

2. Inclusion of All Entities & Relations
- Every entity listed above MUST appear in the generated text.
- Every relation MUST be expressed clearly, connecting the subject and object naturally.
- You may rephrase the relation semantically, but the meaning must remain intact.

3. Integration into Natural Writing
- Blend the entities and relations into a flowing narrative appropriate for the chosen
context type ({context_type}).
- Avoid bullet-point style in the output; it must read like a coherent piece of writing.
- The writing should be creative but faithful to the factual structure provided.

4. No Contradictions or New Visual Details
- Do NOT invent or assign new visual attributes to entities (e.g., do not say "the pole is
red" if not given).
- You may add context, background, or imaginative framing, but it must not contradict the
given information.

5. Optional Creative Expansion
- You may enrich the text with atmosphere, tone, or style fitting the chosen context type.
- Added information must support, not override, the provided facts
```

Figure 7: Textual Context Generation Prompt.

### A.2.3 QUESTION-ANSWER PAIR GENERATION PROMPT

---

**Question-Answer Pair Generation Prompt**

```
You are given a list of structured graph triples sampled from a graph. Each triple is a
JSON object with the keys "subject", "relation", and "object". Your task is to generate a
multi-step question that requires reasoning across ALL the provided triples step-by-step.
The question must not be answerable using only a subset of the triples.

Guidelines:
- The question must not mention any entities that should be inferred. All the intermediate
entities should be inferred step-by-step.
- The final answer to the question must be the last "object" entity in the last triple
- Always mention the image reference in the question if it exists
(e.g., "object in the image").
- Break the question into two sentences if it is too long or complex to keep it clear and
understandable in one sentence. The second sentence should add new context, not repeat
the same information from the first.

Generate a multi-step question and answer, and respond with ONLY a valid JSON object in the
following format:
{{
  "question": "...",
  "answer": "..."
}}

Triples:
[
  {{"subject": "blue", "relation": "is the color of", "object": "cord (Image)"}},
  {{"subject": "cord", "relation": "used during", "object": "event (Product Launch Demo)"}},
  {{"subject": "event (Product Launch Demo)", "relation": "event (Product Launch Demo)
    hosts utility company (Veridian Grid Solutions)", "object": "utility company (Veridian
    Grid Solutions)"}},
  {{"subject": "utility company (Veridian Grid Solutions)", "relation": "utility company
    (Veridian Grid Solutions) maintained by telephone pole", "object": "telephone pole"}},
  {{"subject": "telephone pole (Image)", "relation": "is", "object": "black"}}
]
Notes:
- The answer must be "black" because it is the last object in the last triple.
- The following entities should not be mentioned directly in the question as they
are inferred step-by-step: cord, event (Product Launch Demo), utility company (Veridian
Grid Solutions), telephone pole.
Output:
{{
  "question": "What is the color of the object in the image that maintains the company that
  hosts the event, where the event uses a blue object that is to the left of camera?",
  "answer": "black"
}}

Triples:
[
  {{"subject": "research team (Savanna Ecology Project)", "relation": "research team
    (Savanna Ecology Project) studied giraffe", "object": "giraffe"}},
  {{"subject": "giraffe (Image)", "relation": "is", "object": "walking"}}
]
Notes:
- The answer must be "walking" because it is the last object in the last triple.
- The following entity should not be mentioned directly in the question as it is inferred
step-by-step: giraffe.
Output:
{{
  "question": "What is the entity in the image doing that is studied by the research team
  known as the Savanna Ecology Project?",
  "answer": "walking"
}}

Triples:
[
  {triples}
]
The answer must be "{last_object}" because it is the last object in the last triple.
The following entities should not be mentioned directly in the question as they are inferred
step-by-step: {', '.join(intermediate_objects)}.
Output:
```

Figure 8: Question-Answer Pair Generation Prompt.

### A.2.4 CoT Response Generation Prompt

---

**CoT Response Generation Prompt**

```
You are given a question, its correct answer, and a subgraph that contains the entities and
relations supporting the QA pair.
Your task is to generate a detailed chain-of-thought reasoning output that explains step by
step how the answer follows from the question.

Requirements for the reasoning:
1. Explicitly mention the source of each piece of information:
   - If the evidence comes from an image, say "from image X".
   - If the evidence comes from a figure/table, say "from figure/table Y".
   - If no image or figure/table is involved, assume the information is from the text context
     and say "from the text context".
2. Trace through the relevant entities and relations in the subgraph in logical order.
3. End with the conclusion that matches the provided answer.
4. The reasoning should read naturally, as if another model is thinking through the problem
   step by step.
5. Assume the reader is looking at the images/figures/tables and the text context to answer
   the question.
6. The subgraph is only for reference. The actual reader will not see the subgraph so don't
   generate as if the reader is seeing the subgraph. Don't say anything like "from the
   subgraph", "the relation shows", or "the entity indicates".
7. Do not generate any unnecessary reasoning steps that repeat the same information which is
   already mentioned in previous steps.

Question: What action is performed by individual trained at the institution in the image?
Answer: continues dancing around room
Subgraph: [
{{'subject': 'institution (Central Academy of Contemporary Movement)',
  'object': 'young woman',
  'relation': 'young woman trained under institution
              (Central Academy of Contemporary Movement)'}},
{{'subject': 'young woman',
  'object': None,
  'relation': 'dancing around room',
  'image': 'image 3'}}]

Chain-of-thought reasoning:
The question asks what action is performed by the person trained at the institution.
From the text context, the institution is the Central Academy of Contemporary Movement,
and a young woman trained there.
From image 3, I can see the woman is dancing around the room.
Therefore, the action performed is dancing around room.

Question: What method has lower time cost compared to the another method that is based on an
          algorithm used to obtain the traditional CVT through iterative updates until
          convergence?
Answer: time cost
Subgraph: [
{{'source_entity': "Lloyd's algorithm",
  'target_entity': 'CVT',
  'relationship_description': "The traditional CVT is usually obtained by Lloyd's algorithm,
  iteratively performing updates after each assignment step until convergence is reached."}},
{{'source_entity': 'SLIC',
  'target_entity': "Lloyd's algorithm",
  'relationship_description': "SLIC generates superpixels based on Lloyd's algorithm"}},
{{'source_entity': 'SLIC',
  'target_entity': 'FLIC',
  'relationship_description': 'FLIC's time cost is lower than SLIC's time cost',
  'figure': 'Figure 4'}}]

Chain-of-thought reasoning:
The question asks about what method has lower time cost compared to another method based on
an algorithm for computing the traditional CVT.
From the text context, the traditional CVT is obtained by Lloyd's algorithm, which
iteratively updates until convergence.
From the text context, the method SLIC is based on Lloyd's algorithm.
From Figure 4, it is shown that FLIC's time cost is lower than SLIC's time cost.
Therefore, the method with lower time cost is FLIC.

Question: {question}
Answer: {answer}
Subgraph: {subgraph}
Chain-of-thought reasoning:
```

Figure 9: CoT Response Generation Prompt.

A.2.5 CAPTION-TO-GRAPH CONVERSION PROMPT

---

**Video Caption to Graph Conversion Prompt**

```
You will be given a list of captions (one per scene, in order).
Each caption is paired with a time range.

Your job is to produce BOTH a global entity inventory and per-scene graphs, in two separate
sections.

REQUIREMENTS:

1. Entities section:
- Deduplicate entities across all captions into a single global inventory.
- Assign IDs as stringified integers ("1","2","3").
- Always include "attributes": [].
- Attributes = static/descriptive properties (e.g., "red", "wooden", "wearing hat").
- Do not include transient states (e.g., "sitting", "throwing") here.
- If multiple similar entities are indistinguishable -> group them (e.g., "two dogs").
- If entities are clearly distinct -> create differentiated forms (e.g., "bag_1", "bag_2").
- Do not include interactions with other entities here.
- IMPORTANT: If two entity mentions occur in overlapping or adjacent time ranges, they are
considered coreference *candidates*.
    - Merge them into the same global entity only if the semantics clearly indicate they are
    the same entity
    - (e.g., "former president" at 3s-10s and "man gives a speech" at 5s-12s).

2. Scenes section:
- Each caption corresponds to a scene labeled "Scene N" with its time range (use the given
order, even if time ranges overlap).
- Each scene has a key "relations".
- "relations" is a list of relation triples for that scene.
- Each relation triple must have:
    - "source": source entity ID
    - "target": target entity ID OR null (if no second entity is involved)
    - "relation": a **brief phrase** (not a full sentence) that concisely describes the
    action or interaction
    (e.g., "dog chases cat", "man hands bag to woman", "gives a speech").
- A relation exists if:
    - At least two distinct entities interact, OR
    - A single entity performs an action (then target = null).
- If no valid relations exist for a scene, output "relations": [].

3. General:
- Separate the "entities" section from the "scenes" section.
- Keep relations directional and minimal; avoid redundant inverses.
- Do not invent entities not grounded in captions.

OUTPUT FORMAT:

{{
"entities": [
    {{"id":"1","entity":"ENTITY NAME","attributes":[]}}
],
"scenes": [
    {{
    "scene":"scene_1",
    "relations":[
{{"source":"1","target":"2","relation":"brief phrase describing interaction between entity 1
    and entity 2"}},
{{"source":"1","target":null,"relation":"solo action"}}
    ]
    }}
]
}}

INPUT CAPTIONS:
{annotated}
```

Figure 10: Video Caption to Graph Conversion Prompt.

### A.2.6 GRAPH TRANSFORMATION FOR PARAGRAPH NOT CONTAINING FIGURE REFERENCE PROMPT

---

**Paragraph(without figure reference) to Graph Transformation Prompt**

```
 -Goal-
Given a scientific text and a list of scientific entities, identify ALL relations
expressed in the paragraph.

-Steps-
1. Use the paragraph text only (no figures/tables).
2. Extract any relation between entities that is explicitly stated in the text.
3. For each valid relation, output a JSON object with:
   - source_entity (must come from the provided entities list)
   - target_entity (must come from the provided entities list, or null if not applicable)
   - relationship_description (short, factual, and grounded in the text)
4. Do NOT invent entities that are not in the input entities list.
   Do NOT invent relations and do NOT use prior knowledge.
5. Output must be valid JSON as a list of relation objects (not wrapped in another key).
6. Each relation dictionary must be one line (compact JSON style).
7. Do not include any text outside the JSON.
8. If there are no relations, return an empty list: []

Output Format Example:
[
  {{"source_entity": "Entity1", "target_entity": "Entity2", "relationship_description":
    "Description of relation grounded in text"}},
  {{"source_entity": "Entity3", "target_entity": null, "relationship_description":
    "Another relation grounded in text"}}
]

#####################
Entities:
{entities_json}

Referenced Paragraph:
{paragraph}

#####################
Output (JSON only):
```

Figure 11: Paragraph(with figure reference) to Graph Transformation Prompt.

### A.2.7 GRAPH TRANSFORMATION FOR PARAGRAPH CONTAINING FIGURE REFERENCE PROMPT

---

**Paragraph(with figure reference) to Graph Transformation Prompt**

```
 -Goal-
Given a scientific text (with one or more figure/table captions and references)
and a list of scientific entities, identify ONLY the relations that are
explicitly supported by the figures/tables.

-Steps-
1. Use both the paragraph text and the provided figures/tables.
2. Extract a relation ONLY IF it is directly grounded in the figure or table:
   - Numerical results or metrics reported in the figure/table. Include numerical values if
   available.
   - Explicit comparisons shown in the figure/table (e.g., "X outperformed Y").
   - Visual descriptors of figure elements (e.g., "yellow line corresponds to Model A").
3. Ignore the following completely:
   - Interpretations, hypotheses, or explanations
   (e.g., "improvements are due to larger size").
   - Background details (methods, datasets, phases, tasks, architectures).
4. For each valid relation, output a JSON object with:
   - source_entity (must come from the provided entities list)
   - target_entity (must come from the provided entities list, or null if not applicable)
   - relationship_description (short, factual, and grounded in the figure/table)
   - figure (the figure_label where the relation is supported, never null)
   - idx (list of one or more sentence indices where relation appears, e.g. [0] or [1,2])
5. Do NOT invent entities that are not in the input entities list. Do NOT invent relations
   and do NOT use prior knowledge.
6. Output must be valid JSON as a list of relation objects (not wrapped in another key).
7. Each relation dictionary must be one line (compact JSON style).
8. Do not include any text outside the JSON.
9. If there are no figure/table-grounded relations, return an empty list: []

Output Format Example:
[
  {{"source_entity": "Entity1", "target_entity": "Entity2", "relationship_description":
    "Description of relation grounded in figure/table", "figure": "FigureX", "idx": [0]}},
  {{"source_entity": "Entity3", "target_entity": null, "relationship_description":
    "Another relation grounded in figure/table", "figure": "TableY", "idx": [1,2]}}
]

#####################
Figures:
{figures_json}

Entities:
{entities_json}

Indexed Paragraph Sentences:
{sentences_text}

#####################
Output (JSON only):
```

Figure 12: Paragraph(with figure reference) to Graph Transformation Prompt.

## A.3 QUALITATIVE EXAMPLES OF CRUX BENCHMARK

**Interleaved Image-Text Context**

[ Image 1 ]

... the man—whose name, according to the gym's outdated sign, is "Greg"—is currently engaged in what the Alpine Rescue Academy (Alpine Rescue Academy) once called a controlled challenge. A concept (controlled challenge) so profound, it was designed by designer Elias Truven in 1983, the same year the organization itself was founded. ...

[ Image 2 ]

... And speaking of carabiners—this one, visible in the image, was engineered by Elias Truven himself. A man so precise, he once calibrated a coffee grinder to measure the exact moment a bean should pop. He also advised Mira Tollen on her rope tension algorithms. ...

... The hook, visible in the image, was originally crafted by artisan Mael Tobin—yes, the same Mael Tobin who once made a chandelier out of old bicycle chains and a broken toaster. ...

[ Image 3 ]

... Greg, though we'll never know his real name—trained by the Alpine Rescue Academy. A man who once thought "controlled challenge" meant "don't panic when you're 20 feet up a wall with no idea how to get down." He was told the wall symbolizes concept (controlled challenge), but he just sees it as a wall. ...

[ Image 4 ]

**Question**

What color is the item in the image that is engineered by the designer responsible for the concept symbolized by the structure the person is touching?

**Chain of Thoughts**

The question asks about the color of the item that is engineered by the designer responsible for the concept symbolized by the structure the person is touching.

1 From image 1, the man is touching climbing wall.

2 From text context, climbing wall symbolizes concept(controlled challenge).

3 From text context, concept(controlled challenge) is designed by designer Elias Truven.

4 From text context, carabiners were engineered by Elias Truven himself.

5 From image 2, color of the carabiner is silver.

**Answer**

Silver

Figure 13: CRUX example from video source.

**Interleaved Image-Text Context**

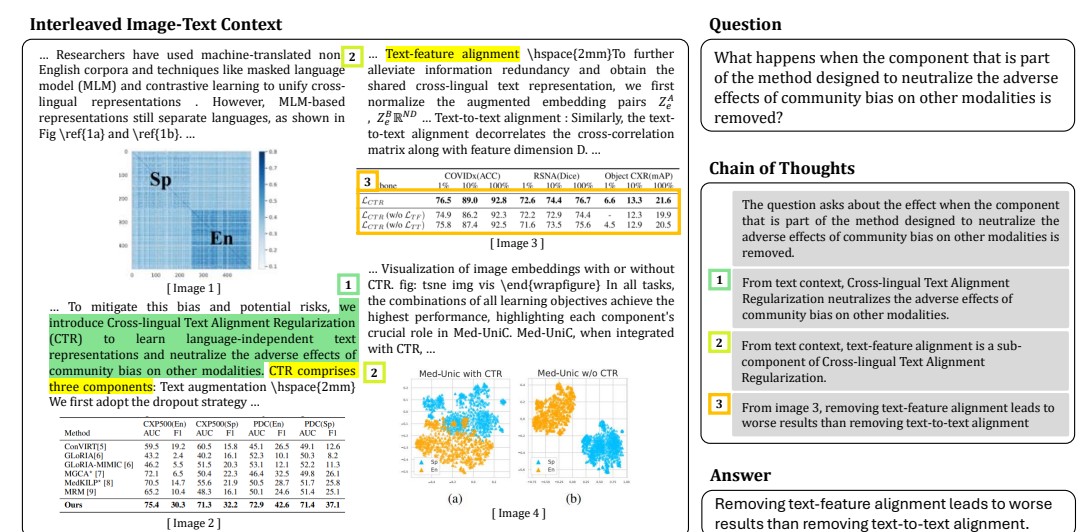

**Question**

What happens when the component that is part of the method designed to neutralize the adverse effects of community bias on other modalities is removed?

**Chain of Thoughts**

The question asks about the effect when the component that is part of the method designed to neutralize the adverse effects of community bias on other modalities is removed.

1 From text context, Cross-lingual Text Alignment Regularization neutralizes the adverse effects of community bias on other modalities.

2 From text context, text-feature alignment is a sub-component of Cross-lingual Text Alignment Regularization.

3 From image 3, removing text-feature alignment leads to worse results than removing text-to-text alignment

**Answer**

Removing text-feature alignment leads to worse results than removing text-to-text alignment.

Figure 14: CRUX example from scientific paper source.

