# OpenReview forum: "Getting to the CRUX: Graph-Based Data Generation for Advancing Multi-Hop Cross-Modal Reasoning"
_ICLR.cc/2026/Conference — ICLR 2026 Conference Withdrawn Submission_

### Official Review · Reviewer_hmjU · 2025-10-22

**Soundness:** 2
**Presentation:** 2
**Contribution:** 2
**Rating:** 4
**Confidence:** 3

**Summary:**

The paper introduces CRUX, a graph-based data generation pipeline and a benchmark for cross-modal multi-hop reasoning. The authors utilized the images with scene graphs annotations and leveraged LLM to generate entities with relationships, textual context and QA pairs. The benchmark consists of three categories: natural images, videos and scientific papers from GQA, and ActivityNet Captions SPIQA datasets respectively. The experiment results demonstrate that finetuned models on CRUX perform better on manually verified CRUX test set and other benchmarks.

**Strengths:**

•	The creation pipeline of the dataset is explained very clearly and in detail.

•	The motivation and limitations of current approaches for the dataset generation pipeline are clearly stated.

•	Test set is manually verified for reliable evaluation.

**Weaknesses:**

•	The questions are complex to understand, and the details of the text seem unrelated. Although it follows the multi-hop reasoning across image and text, the reviewer wonders how humans perform on CRUX. If the authors conduct a human evaluation test, the results show how the dataset aligns with human performance.

•	The authors did not give details on the current multi-hop datasets in the related work section. To understand the novelty in the process, comparing it with other datasets is helpful. How is CRUX distinguished from current benchmark datasets?

•	The experiment section lacks the clarity found in the rest of the paper. It could be improved by highlighting the main takeaways more explicitly.

**Questions:**

•	What is image-to-image reasoning mentioned in Section 4.2? The CRUX focuses on cross-modal multi-hop reasoning. Then how is it related to the focused area?

•	What is the reason to finetune the model on both Mantis-Instruct and CRUX?

•	Table 3 shows performance results of Idefics2 models trained on different datasets. The finetuned model on both Mantis-Instruction and CRUX performs better on SPIQA datasets. However, the SPIQA data has already been used to generate questions in CRUX under the guise of scientific papers. Can the authors clarify how they ensured that this data did not positively influence the model’s performance on SPIQA?

•	The paper has duplicate text in Section 5.3, which explains the dataset twice.

•	CRUX focuses on multihop reasoning across modalities. Why did you not compare your results with other multihop benchmarks instead of comparing the performance on multi-image and multi-text datasets in Section 5.3?

•	Section 5.4 is not clearly explained. What is the source for the images? Is the same pipeline used to augment the data?

•	The authors did not add the section for “Using LLM”.

---

### Official Review · Reviewer_RhS2 · 2025-10-28

**Soundness:** 3
**Presentation:** 2
**Contribution:** 3
**Rating:** 4
**Confidence:** 4

**Summary:**

This paper introduces a new image-text question answering benchmark, CRUX, which is the first systematic cross-modal multi-hop reasoning dataset built on graph data, spanning images, videos, and scientific papers. The experimental results demonstrate that models fine-tuned on CRUX achieve significant performance improvements across tasks in three domains, as shown in Table 2. Moreover, the results in Tables 3 and 4 highlight the enhanced generalization ability gained by incorporating CRUX into other reasoning benchmarks. Finally, the authors further show the effectiveness of augmenting CRUX on the tasks of natural images, as presented in Table 5.

**Strengths:**

1. This work presents the first graph-based, cross-modal, multi-hop reasoning benchmark for vision-language models (VLMs), spanning diverse modalities including natural images, videos, and scientific papers. The proposed automatic data generation pipeline effectively leverages graph structures as a reliable source for multi-modal content synthesis.

2. The empirical results across Tables 2–5 clearly demonstrate the effectiveness and broad applicability of the proposed benchmark in evaluating multi-hop multi-modal reasoning capabilities.

3. The authors further propose a novel metric, Reference Accuracy, which measures whether a model correctly identifies the necessary visual evidence to answer a question. This metric serves as an important indicator of whether the model performs reasoning grounded in the corresponding visual content, rather than relying on memorization or guesswork, an essential aspect for assessing multi-hop cross-modal reasoning.

**Weaknesses:**

1. **Discussion on Number of Hops**. This work focuses on multi-hop cross-modal reasoning, yet it provides limited discussion on the number of hops involved. For instance, the data statistics in Table 1 do not include the average number of hops, even though Section 3.3 mentions that $1 \leq h \leq 5$. Moreover, the experimental section does not analyze model performance across different hop counts. As a benchmark paper, this omission represents a significant gap in evaluating the dataset’s multi-hop reasoning capability.

2. **Effectiveness of Reference Accuracy Metric**. While I appreciate the introduction of the Reference Accuracy metric, the inclusion of CRUX does not seem to yield substantial improvements in this metric for video and text-dense domains, as shown in the row of $\text{Qwen2.5-VL}{\text{CRUX}}$. Furthermore, considering that this model is specifically finetuned for the proposed benchmark, the large gap in Reference Accuracy between $\text{Qwen2.5-VL}{\text{CRUX}}$ and GPT-4o (44.3% vs. 72.2% and 40.8% vs. 56.5%) appears unacceptable.

3. **Choice of Model for Finetuning**. I also noticed that the authors report the zero-shot performance of Qwen2.5-VL (72B), which already achieves strong results in Reference Accuracy. This raises the question of why the authors chose to finetune on this particular model. If computational resources were the main constraint, finetuning on a slightly larger model (e.g., 13B or 32B) could better demonstrate the scalability and effectiveness of CRUX on larger architectures. It might also help achieve comparable or superior Reference Accuracy on video and text-dense tasks compared to proprietary models.

**Questions:**

1. The main issues have been discussed above. If the authors can address these weaknesses, I would consider improving my score.

2. Could the authors explain why the CoT-based approach outperforms the non-CoT variant in Table 3?

3. Why did the authors adopt specific data filtering strategies for the training data? While I appreciate the human verification conducted on the test set, it is recommended to consider similar measures during training as well, since CRUX uses an LLM to extract entities and relationships from graphs, which may introduce hallucinated or noisy content.

---

### Official Review · Reviewer_nrd9 · 2025-10-28

**Soundness:** 3
**Presentation:** 3
**Contribution:** 3
**Rating:** 4
**Confidence:** 4

**Summary:**

The paper proposes a new benchmark for multi-hop multi-modal reasoning, constructed based on graphs such as scene graphs. It then leverages LLMs to generate relationships between entities, textual contexts, and question-answer pairs. The authors evaluate multiple VLMs on this benchmark and show that they perform poorly, while fine-tuning on the provided training set leads to performance improvements.

**Strengths:**

1. The paper is well-written and easy to follow.
2. The authors provide both training and test sets and demonstrate that fine-tuning on their training set leads to notably improvments in multi-hop multi-modal reasoning.

**Weaknesses:**

1. A more detailed analysis of the benchmark is needed. For example,
- Distribution of reasoning hops
- Number of samples per hop
- Model accuracy across hops
- Distribution of reasoning types (e.g., percentage of visual vs. textual reasoning)

2. The proposed pipeline appears to rely on the availability of graph information (e.g., scene-graph) from the source data, which suggests that it may not be applicable to datasets lacking such graph annotations.

**Questions:**

Missing relevant work citation:
 - II-MMR: Identifying and Improving Multi-modal Multi-hop Reasoning in Visual Question Answering, ACL'24

---

### Official Review · Reviewer_kbGw · 2025-10-30

**Soundness:** 3
**Presentation:** 3
**Contribution:** 2
**Rating:** 4
**Confidence:** 4

**Summary:**

The paper introduces CRUX, a multimodal reasoning dataset featuring enforced multi-hop dependency among textual and visual objects, which is achieved by generating data with structured graphs. Experimental results show that current VLMs struggle to solve the proposed task, while fine-tuning with CRUX dataset can improve the results both on its test set and on other visual reasoning benchmarks. The contributions include a benchmark with training data, and a pipeline to generate this data from scene graphs and other sources.

**Strengths:**

1. This work proposes an effective methodology to collect multimodal reasoning data with complex relations and groundings. The graph-based design ensure that the dependencies are necessary for answering the question.
2. The paper is well-organized and clearly presented.
3. Experimental results show the benefit of using the contributed dataset to improve performances on multi-hop reasoning tasks.

**Weaknesses:**

1. The question style in the proposed benchmark looks too "synthetic" (i.e., involves too many clauses, relations and entities), which does not match the natural style of real-world multi-hop questions. Thus, it is not surprising that current VLMs achieve low scores, and fine-tuning yields significant improvement (Table 2). In contrast, Table 4 is more important for measuring the contribution of CRUX as an auxiliary task, but it is unclear whether the marginal improvements comes from domain similarity or just larger amount of data involved. It may be necessary to add some ablation experiments or case studies to clarify that the advantage comes from the subclass of questions involving multi-hop reasoning.
2. Novelty is limited as it is widely-recognized that LLMs nowadays are capable for synthesizing tasks with complex data structures.

**Questions:**

1. As mentioned above, could you provide more evidence that the improvements in Table 4 originate from the improvement of multi-hop reasoning capabilities instead of domain similarity or just larger data amount?
2. In Table 3, some CoT models underperform their no-CoT version. Does this indicate that explicit multi-hop reasoning can actually be harmful? How could we avoid this?

---

### Note · Authors · 2025-11-14

I have read and agree with the venue's withdrawal policy on behalf of myself and my co-authors.